# Bioinformatic Workflows for Generating Complete Plastid Genome Sequences—An Example from *Cabomba* (Cabombaceae) in the Context of the Phylogenomic Analysis of the Water-Lily Clade

**DOI:** 10.3390/life8030025

**Published:** 2018-06-21

**Authors:** Michael Gruenstaeudl, Nico Gerschler, Thomas Borsch

**Affiliations:** 1Institut für Biologie, Systematische Botanik und Pflanzengeographie, Freie Universität Berlin, 14195 Berlin, Germany; nic94@zedat.fu-berlin.de (N.G.); t.borsch@bgbm.org (T.B.); 2Botanischer Garten und Botanisches Museum Berlin, Freie Universität Berlin, 14195 Berlin, Germany; 3Berlin Center for Genomics in Biodiversity Research (BeGenDiv), 14195 Berlin, Germany

**Keywords:** bioinformatics, *Cabomba*, genome assembly, phylogenomics, plastid genome, standardization, workflow

## Abstract

The sequencing and comparison of plastid genomes are becoming a standard method in plant genomics, and many researchers are using this approach to infer plant phylogenetic relationships. Due to the widespread availability of next-generation sequencing, plastid genome sequences are being generated at breakneck pace. This trend towards massive sequencing of plastid genomes highlights the need for standardized bioinformatic workflows. In particular, documentation and dissemination of the details of genome assembly, annotation, alignment and phylogenetic tree inference are needed, as these processes are highly sensitive to the choice of software and the precise settings used. Here, we present the procedure and results of sequencing, assembling, annotating and quality-checking of three complete plastid genomes of the aquatic plant genus *Cabomba* as well as subsequent gene alignment and phylogenetic tree inference. We accompany our findings by a detailed description of the bioinformatic workflow employed. Importantly, we share a total of eleven software scripts for each of these bioinformatic processes, enabling other researchers to evaluate and replicate our analyses step by step. The results of our analyses illustrate that the plastid genomes of *Cabomba* are highly conserved in both structure and gene content.

## 1. Introduction

The sequencing of plastid genomes constitutes a reliable and well-tested method in plant genomic research. For example, sequencing plastid genomes for phylogenetic purposes (commonly referred to as “plastid phylogenomics”) has been conducted since the mid-2000s (see [1] for a review). Over the past five years, plastid phylogenomics has been employed to infer the origin of land plants [2,3], the evolutionary history of major orders of flowering plants [4,5,6] and the phylogenetic relationships within various plant families [7,8]. Plastid genome sequences have also been used to infer phylogenetic relationships among genera [9,10] and species [11,12,13], to resolve recently radiated plant lineages [14] and to evaluate valuable herbarium specimens [15].

Several factors render the comparison of plastid genomes beneficial for phylogenetic analysis. First, the majority of plastid genomes display strong structural conservation in both gene composition and synteny, enabling the comparison of primarily homologous regions across a wide variety of taxa [16]. Second, plastid genomes are typically uniparentally (maternally) inherited and display a protected form of recombination [17], thus avoiding conflicting phylogenetic signals generated through incomplete lineage sorting or hybridization. Third, plastid genomes occur in high copy numbers per plant cell, allowing easy isolation and DNA sequencing even from old and degraded samples [18]. Fourth, given the mosaic-like and idiosyncratic mutational dynamics of different plastid genome regions such as genes, introns, and spacers [19], the comparison of plastid genomes can help to infer phylogenetic relationships at various taxonomic levels [20]. Fifth, the comparison of plastid genomes, especially complete genome sequences, enables the identification of rare evolutionary events, such as inversions or translocations of entire genome regions [21,22]. As a consequence, some authors have suggested the use of complete plastid genomes as potential DNA barcodes for plant identification [23,24].

Improvements in DNA sequencing technology have rendered the sequencing of complete plastid genomes highly efficient. Next-generation sequencing (NGS) enables the quick generation of large amounts of sequence data through a parallelization of the sequencing process [25]. Contemporary NGS machines generate billions of nucleotides per machine run and have become widely available [26,27]. The ubiquity of NGS in biological research has led to an unprecedented accumulation of DNA sequence data [28]. In particular, NGS-driven data generation sparked a surge of publicly available plastid genome sequences: While approximately 160 plastid genomes were available in the nucleotide section of NCBI (https://www.ncbi.nlm.nih.gov/) in fall 2009 [1], 1000 such genomes were available in spring 2016 [29]; as of fall 2017, more than 4100 plastid genome sequences had been deposited to NCBI. This quadrupling within one and a half years suggests that many more plastid genomes will be sequenced and submitted over the coming years [30].

The trend towards massive sequencing of complete plastid genomes highlights the need for standardized and well-documented bioinformatic workflows in plastid genomics, especially for genome assembly, annotation, sequence alignment and phylogenetic inference. The current surge of plastid genome sequences in public sequence databases offers tremendous opportunities for molecular and evolutionary research but also poses new challenges for correct data analysis. The need for efficient and standardized bioinformatic workflows that allow reproducibility of the precise genome sequences is particularly apparent [31]. Moreover, many researchers fail to provide the necessary details to replicate their bioinformatic analyses and often merely list name and version number of the software tools utilized. For example, a recent paper in this journal reported the results of sequencing and comparing complete plastid genomes of the genus *Ilex* [32], but did not provide information on contig assembly beyond stating that a “combination of de novo and reference-guided assembly strategies were performed using Celera Assembler 6.1, Newbler 2.6 and Newbler 2.8 assemblers (Roche)” [32] (p. 3). Without detailing the order and settings under which these assembly software tools were employed, contig assembly and, by extension, genome assembly cannot be evaluated.

A strong emphasis is being placed on the standardization and reproducibility of bioinformatic workflows in NGS-driven genomic research [33,34]. This emphasis is motivated by the experience that in many multi-step bioinformatic analyses, the precise settings of individual software tools can affect downstream analyses and, ultimately, the final results [35]. Phylogenomic analyses are a prime example for this sensitivity to specific analytical settings. Magoc et al. [36], for example, report that the outcome of the assembly of bacterial genomes is strongly dependent on the choice of assembly software. Similarly, Morrison et al. [37] report that the choice of assembly and annotation software can have a significant impact on the final product and recommended that tracking the “analytic provenance” of the assembly process is critical for a reproducible analysis. Nekrutenko and Taylor [33] and Kanwal et al. [38] list further examples of non-reproducibility of genomic research related to the choice and usage of analysis software. These investigations agree that the software settings and parameter values used for contig and genome assembly must be co-supplied with a final genome sequence if proper evaluation and reproducibility are to be achieved.

In the present investigation, we report the laboratory procedure, the bioinformatic workflow and the results of sequencing and characterizing three novel and complete plastid genomes. Specifically, we provide a detailed description of the bioinformatic steps taken to assemble and annotate three complete plastid genomes of the aquatic plant genus *Cabomba* (Cabombaceae). Upon assembly and annotation of the three genomes, we compare them against each other and with the previously published plastid genome of *C. caroliniana* [6]. Thereafter, we extract and align the coding regions of all four genomes to provide an interface to a subsequent phylogenetic analysis. Coding regions are favored over non-coding regions here as in many plastid phylogenomic investigations due to their reduced need for manual correction upon sequence alignment [4,7,8]. Finally, we conduct a simple phylogenetic tree inference as a proof of concept for the completeness of our workflow.

The genus *Cabomba* comprises five species of perennial aquatic plants that naturally occur in tropical and subtropical freshwater in the Americas, particularly in the southern United States, Mexico, Central and South America [39,40]. One of the species of *Cabomba*, *C. caroliniana*, has spread to aquatic ecosystems across the globe [41] and is considered a highly invasive species in many regions of the world [42,43]. Moreover, *Cabomba* is one of two genera of the plant family Cabombaceae, which belongs to one of the earliest diverging lineages of the flowering plants [6]. The latter study also indicated that the precise relationships within the waterlily clade (Nymphaeales) are not fully understood. Thus, knowledge of the complete plastid genomes of species of *Cabomba* is both of ecological and evolutionary interest.

Central to this investigation is a detailed description of the bioinformatic steps for assembly and annotation as well as the extraction and alignment of coding regions of complete plastid genomes. Specifically, we present and discuss custom software scripts that we developed and applied for these bioinformatic processes as well as for generating assembly quality statistics. Our description also includes a tutorial which illustrates the application of each script as part of a semi-automated workflow and is sufficiently detailed to allow other researchers to evaluate and replicate our analyses. The software scripts are available as Appendix A to this article and are additionally maintained in a public repository at the code sharing platform GitHub (https://github.com/michaelgruenstaeudl/StandardizedPlastidPhylogenomics). In addition, the full set of quality-filtered reads, as well as the raw annotation files for *C. aquatica*, are available from Zenodo (https://zenodo.org/record/1213269).

## 2. Materials and Methods

### 2.1. Plant Material and Taxon Sampling

The complete plastid genomes of three taxa of the genus *Cabomba* were newly sequenced for this investigation: *C. aquatica*, *C. caroliniana* cv. ‘Silvergreen’ and *C. furcata*. Individuals of all three species were cultivated at the Botanical Garden and Botanical Museum Berlin. The taxonomic identity of each species was confirmed against the descriptions and the species key of [39]. During taxonomic confirmation, the sample of *C. caroliniana* was found to display twisted leaf segments and a silver-green leaf color, which is a combination of features not consistent with any of the subspecies listed by [39]. This entity is traded as an aquatic ornamental plant and has been called *C. caroliniana* ‘Silvergreen’ to highlight its silver-green leaf color. However, its precise status from an evolutionary perspective is unclear. A complete list of species names, herbarium vouchers and GenBank accession number of each taxon analyzed is given in Table 1.

### 2.2. DNA Isolation, Shearing and Size Selection

Total genomic DNA was extracted from young leaf material after a custom cleaning procedure to remove any organisms growing on the leaf surfaces. During cleaning, all leaf sections covered with algae were removed with sterile razor blades, followed by two rinsing steps of the shaved leaves with de-ionized water and 70% ethanol. The leaf material so cleaned was dried on silica gel, and total genomic DNA was isolated from 25 mg of dried material with the NucleoSpin plant II kit (Macherey-Nagel, Düren, Germany). The isolated DNA was sheared to an average fragment size of 550 bp with the Covaris S220 sonicator (Covaris, Woburn, MA, USA) under the following settings: 5% duty factor, 175 W peak incident power, 200 cycles per burst, frequency sweeping and a duration of 70 s. After sonication, DNA fragments of the size range 400–900 bp were extracted on a 1.5% agarose gel using the BluePippin standard (Sage Science, Beverly, MA, USA). To confirm the successful size selection of DNA fragments prior to library preparation, quantity and length distribution of the DNA fragments were evaluated with the Agilent 2100 Bioanalyzer (Agilent Technologies, Santa Clara, CA, USA). All laboratory steps were conducted following the manufacturers’ instructions.

### 2.3. Library Preparation and DNA Sequencing

For each sample, a barcoded genomic library was constructed using the Illumina TruSeq DNA sample preparation kit (Illumina, San Diego, CA, USA) following the low sample protocol in the manufacturer’s instructions. Standard indexing adapters, one for each sample, were hereby ligated to the ends of the DNA fragments to generate single-indexed libraries. Finished libraries were quantified with the Agilent 2100 Bioanalyzer. To confirm the success of adapter-ligation and to estimate optimal pooling ratios, a qPCR was conducted for each sample on a Mastercycler ep realplex (Eppendorf AG, Hamburg, Germany) using the KAPA library quantification kit (*KAPA Biosystems*, Wilmington, MA, USA). Upon qPCR, the indexed DNA libraries were normalized to a concentration of 2 nM and pooled in equal volumes. Pooled libraries were sequenced as paired-end reads on an Illumina MiSeq platform (Illumina, San Diego, CA, USA) with v3 chemistry, 600 cycles and an insert size of 250–300 bp. Sequencing was conducted at the Berlin Center for Genomics in Biodiversity Research.

### 2.4. Genome Assembly

#### 2.4.1. Generating Ordered Intersection of Paired-End Reads

Upon sequencing, the raw reads of each sample were filtered to retain only those sequences that contained both R1 and R2 reads (i.e., the intersection of the two read sets). This filtering ensures that orphaned reads were removed prior to quality filtering and genome assembly. Filtered reads were then sorted by read header using bioawk v.20110810 [44], ensuring that R1 and R2 reads of each sequence are in matching position in the read files. The number of reads is counted before and after generating the ordered intersection to quantify the read loss during this procedure. All steps were automated through a custom Bash [45] script, which is given as script 1 (Appendix A).

#### 2.4.2. Quality Filtering of Raw Reads

Upon generating the ordered intersection of paired-end reads for each sample, the reads of each sample were filtered and trimmed by quality using FASTX Toolkit v.0.0.14 [46] and a minimum quality score of 30. The number of reads was counted before and after quality filtering to quantify the read loss during this procedure. Upon quality filtering, Illumina adapters, if present, were trimmed from the reads using module ‘BBduk’ of BBtools v.33.89 [47]. All steps of the quality filtering process were automated through a custom Bash script, which is given as script 2 (Appendix A).

#### 2.4.3. Assembly of Contigs

Trimmed and quality-filtered reads were assembled de novo into contigs using a three-step procedure, which was automated through the IOGA pipeline v.20160910 [15]. First, the reads were assembled into preliminary contigs using SOAPdenovo2 v.2.04 [48], testing a range of kmer values to optimize contig length (kmer = 33–97, in increments of 4). Second, the preliminary contigs were used as reference sequences to enrich target-specific reads from all quality-filtered reads using module ‘BBmap’ of BBtools. Third, final contigs were assembled from the enriched read set using SPAdes v.3.1.1 [49], testing four different kmer values (k = 33, 55, 77 and 95) to optimize contig length. The chloroplast origin of final contigs was confirmed by IOGA through the mapping of contigs against a reference database comprising the complete plastid genomes of *Cabomba caroliniana* and *Brasenia schreberi* [6] using module ‘BBmap’ of BBtools; all contigs that mapped to any of the database genomes were hereby retained. The number of, and the average read coverage across, the final contigs was automatically extracted from the IOGA output to quantify assembly success. All steps of the assembly process were automated through a custom Bash script, which is given as script 3 (Appendix A).

#### 2.4.4. Manual Assembly of Contigs

Individual contigs were combined manually into final assemblies with Geneious v.10.2.3 [50], using the previously published plastid genome of *Cabomba caroliniana* as a reference for correct contig position and orientation. Initial positioning was hereby achieved through pairwise alignment of the contigs against the reference genome with MAFFT v7.309 [51], using a gap open penalty of 1.23 and the setting to automatically determine sequence direction and best alignment algorithm. Upon correct positioning, individual contigs were stitched together if their 5′ or 3′ ends shared an overlap of at least 50 nucleotides and less than 10% nucleotide dissimilarity. This procedure resulted in the assembly of near-complete plastid genome sequences, missing only the second inverted repeat (IR) regions.

#### 2.4.5. Identification and Confirmation of IR Boundaries

Start and end position of the IR region on a genome assembly that, at this point, misses the second IR were identified through the automatic mapping of all quality-filtered reads against this assembly using bowtie2 v.2.3.2 [52]. If only one IR is present in the assembly, it receives twice as many mapping hits on average as the single copy regions. For each genome, the region so identified was compared to the final contigs generated via the automatic assembly and, in the case of an exact match, appended as a reverse complement to the genome assembly, thereby completing the assembly. The quadripartite structure of the final assembly and the equality of the inverted repeats were confirmed by blasting each final assembly against itself using the BLAST+ suite v.2.4.0 [53]. Any nucleotide ambiguities in the final assembly were resolved by mapping the quality-filtered reads against regions containing such ambiguities using bowtie2 and correcting them based on the majority-rule consensus across the mapped reads. Self-blasting was automated through a custom Bash script, which is given as script 4 (Appendix A).

#### 2.4.6. Extraction of Plastid Reads

All quality-filtered reads used for plastid genome assembly (i.e., plastid reads) were extracted from the read files using a combination of samtools v.1.7 [54] and BEDtools v.2.27.1 [55] upon mapping the quality-filtered reads against the final assembly of each sample using bowtie2. All steps of this mapping and extraction procedure as well as the compilation of mapping statistics were automated through a custom Bash script, which is given as script 5 (Appendix A).

#### 2.4.7. Assembly Quality Statistics and Genome Maps

Upon the extraction of plastid reads, a series of statistics was calculated to evaluate the assembly quality for each sample. These quality statistics comprised (a) the number and proportion of quality-filtered reads that mapped to the final assembly; (b) the mean read length; (c) the mean coverage depth, and (d) the number of nucleotide positions of a final assembly with a coverage depth equal or greater than 20-, 50- and 100-fold. All steps of generating these statistics were automated through a custom Bash script, which is given as script 6 (Appendix A). In addition to assembly quality statistics, contig statistics of the newly-generated plastid genomes were calculated with QUAST v.5.0 [56]. Circular genome maps were generated via OGDRAW v.1.2 [57] (Figure 1 and Appendix A).

### 2.5. Genome Annotation

The complete plastid genomes of *Cabomba* were annotated in a two-step procedure.

#### 2.5.1. Generating Raw Gene Annotations

First, raw gene annotations were generated based on the predictions of the annotation servers DOGMA [58] and cpGAVAS [59], selecting the plastid genomes of *Nuphar advena* [60] and *Nymphaea alba* [61] as references. The annotations from DOGMA were then converted from plain text to GFF format [62] using a custom Bash script, which is given as script 7 (Appendix A). The annotations generated by both servers can differ in aspects such as precise gene length and position, requiring inspection by eye upon. Thus, the annotations of both servers were loaded into Geneious, combined into a single annotation track, and saved as a single file in GFF format. The union of both annotations sets was then generated with a custom Bash script, which is given as script 8 (Appendix A). The script integrates code for, and interacts with, the interpreters of Python [63] and Perl [64] and was tested with Python v.2.7.14 and Perl v.5.26.1. After generating this union annotation set, its annotations were imported back into Geneious.

#### 2.5.2. Manual Correction of Annotations

Second, all annotations were inspected by eye and, where necessary, corrected manually using Geneious. During this inspection, we confirmed (a) the presence of terminal start and stop codons for each coding region, (b) the absence of any internal stop codons, and (c) that the length of each coding region was a multiple of three. In cases where no terminal start and stop codons were encountered, regions up to 50 bp upstream and downstream of the respective coding region were inspected for alternative start and stop codons; the annotation under study was extended accordingly. In cases where internal stop codons were encountered in genes with an intron–exon structure, the start and stop positions of the individual exon regions were re-evaluated. In cases where RNA editing of the start codon was suspected to precede translation initiation [6], a feature exception specifying the details of the editing was appended to the gene annotation. Upon manual correction of all annotations, the complete and annotated genome sequences of the newly sequenced plastid genomes were submitted to GenBank (Table 1).

### 2.6. Sequence Alignment

Upon curation and finalization of their gene annotations, the three newly sequenced as well as the previously published plastid genome of *Cabomba* were saved as individual files in GenBank format. Using these files as input, the coding regions of all four plastid genomes were extracted, aligned and post-processed using a two-step procedure.

#### 2.6.1. Extraction and Alignment of Coding Regions

The coding regions were extracted and grouped by gene name, translated from nucleotides to amino acids, aligned based on their amino acid sequences using MAFFT under default settings (gap open penalty of 1.23, automatic determination of sequence direction and best alignment algorithm), back-translated to nucleotide sequences, and concatenated as gene-wise nucleotide alignments. The back-translation was conducted with the help of script ‘align_back_trans’ of the Galaxy ToolShed [65]. Conducting alignments on amino acid instead of nucleotide sequences has the advantage of ensuring that any insertion or deletion within a coding region constitutes a multiple of three and that the trinucleotide reading-frame of codons is maintained. In addition, conducting alignments on gene-wise instead of genome-wide sequences has the advantage of preventing any insertion or deletion at the beginning or end of a coding region to cause an overlap of different genes in the resulting alignment. The alignment so generated constitutes a concatenation of aligned coding regions and was saved as a NEXUS file. All steps of extracting, translating and aligning coding regions from the input genomes were automated through a custom Python script, which is given as script 9 (Appendix A).

#### 2.6.2. Removal of Gap Positions

Upon alignment and concatenation of coding regions, gap positions in the resulting alignment were identified and excluded using the trimAl package v1.4.rev22 [66]. In addition, alignment statistics were calculated before and after gap removal. Since the input alignment represents a concatenation of coding regions, the removal of gap positions is important because the inadvertent absence of an annotation for a gene or an exon would otherwise be interpreted as a deletion in the matrix used for phylogeny inference. The gap-free alignment was saved as a FASTA file. All steps of the gap removal and the calculation of alignment statistics were automated through a custom Bash script, which is given as script 10 (Appendix A).

### 2.7. Phylogenetic Analysis

As a proof of concept, the phylogenetic relationships between the three newly sequenced as well as the previously published plastid genome of *Cabomba* were inferred from the gap-free alignment under the maximum likelihood (ML) optimality criterion and the GTR + I + G nucleotide substitution model in R v.3.4.4 [67] using the R-package ‘phangorn’ v.2.4.0 [68]. Node support for the best ML tree was calculated via 1000 bootstrap (BS) replicates under the same settings. All steps of the phylogenetic inference process were automated through a custom R script, which is given as script 11 (Appendix A). Upon tree inference, the script saves the phylogenetic tree with the highest likelihood score, the corresponding bootstrap values and several parameter estimates as publication-ready vector graphic as well as NEXUS-formatted file for further analysis.

### 2.8. Overview of Workflow and Tutorial

A step-by-step overview of our bioinformatic workflow, the individual scripts and software tools utilized and the average computation time of each automated step is presented in Table 2. An overview of the command-line instructions necessary to initiate and carry out the workflow is provided in a tutorial (Appendix A).

## 3. Results

### 3.1. Genome Assembly

Between 1.9 and 4.1 million read pairs were generated per sample (Table 3). Of these, between 1.12% and 3.87% were of chloroplast origin upon quality filtering. The proportion of plastid reads was not correlated with the number of total reads generated per sample. For each newly-generated plastid genome, a mean coverage depth greater than 200 was measured, with more than 99.7% of all bases covered by a depth of 50 or greater. Between three and four contigs with a length equal or greater than 1000 bp were generated by the automatic assembly process, with the largest contig matching the large single-copy (LSC) region in each sample. For *C. aquatica* and *C. furcata*, the three contigs equal or greater than 1000 bp exactly matched the LSC, the small single-copy (SSC) and the IR region, greatly reducing the complexity of the manual contig stitching process. The contiguity statistics N50 and L50 also indicated high quality contigs for all three plastid genomes, indicating that a single contig covers at least half of the assembly. Regarding computation time, de novo assembly of contigs via IOGA was on average the most time-consuming automated step of our bioinformatic workflow, followed by the filtering of reads by quality score (Table 2).

### 3.2. Genome Structure and IR Length

The three newly sequenced plastid genomes of *Cabomba* were found to display the default genome structure of land plant plastid genomes: Each of them displays a quadripartite structure, with the IR regions separating the LSC from the SSC (Figure 1). The complete length of all plastid genomes under study ranges from 159,487 bp in *C. aquatica* to 164,057 bp in *C. caroliniana* (Table 4). The length variation of the three structural regions hereby differs slightly, with a notable increase of the length of the IR region, and thus the number of genes contained in them, in the previously sequenced plastid genome of *C. caroliniana*. Specifically, the IR of this genome was found to be more than 6 kbp larger than the IR of all other genomes analyzed, including the plastid genome of its presumed conspecific relative *C. caroliniana* cv. ‘Silvergreen’. The proportion of coding to non-coding regions and the guanine and cytosine (GC) content of the genome was highly conserved across the study taxa. Genome maps of each newly sequenced plastid genome are available in Figure 1 and Appendix A, respectively.

### 3.3. Gene Content

The gene content of the three newly sequenced plastid genomes of *Cabomba* was found to be highly conserved (Table 4). All plastid genomes under study display a set of 116 unique genes, of which either 9 (or 19 for the previously sequenced plastid genome of *C. caroliniana*) are duplicated in the IR regions. Within this gene complement, 82 are protein-coding genes, 30 are transfer RNAs (tRNAs), and four are ribosomal RNAs (rRNA). Each plastid genome under study displays a trans-spliced version of the gene *rps*12, which comprises three exons; the first exon of *rps*12 is located in the LSC, while the second and third exons are located (and thus duplicated) in the IR regions.

### 3.4. Alignment Statistics

Prior to the removal of gap positions, the concatenation of aligned coding regions across all samples of *Cabomba* displayed a length of 68,922 bp, with an average pairwise sequence identity of 0.9852. After the removal of gap positions, the total alignment length decreased to 68,451 bp, with an average pairwise sequence identity of 0.9891. For the pair of the most similar sequences (i.e., the two sequences of *C. caroliniana*), the pairwise sequence identity also increased through gap removal, reaching a score of 0.9998, which represents a total of 12 nucleotide changes across all coding regions.

### 3.5. Phylogenetic Inference

The results of our phylogenetic inference indicate that the two samples of *C. caroliniana* are more closely related to each other than to any other *Cabomba* species under study (Figure 2). The split between the clade formed by *C. caroliniana* and the two other *Cabomba* species is supported by full bootstrap support (BS 100%). The comparatively long terminal branch of *C. aquatica* is indicative of several autapomorphic nucleotide changes in this taxon, which are also visible by manual inspection of the alignment.

## 4. Discussion

In the present investigation, we outlined the sequencing, assembly, annotation, gene alignment and phylogenetic inference process for three complete plastid genomes of the aquatic plant genus *Cabomba*. We found that the plastid genomes of *Cabomba* are highly conserved in both structure and gene content. At the same time, we found a peculiar difference in the IR length between a newly-sequenced cultivar of *C. caroliniana* and a previously published plastid genome of the same species. While changes in IR length have been identified during the evolution of many plant lineages [69], expansions or contractions of the IR within the same species are unusual. One way to explain this observation is the presence of a greater genetic diversity within *C. caroliniana* than currently known. This hypothesis is supported by the results of McCracken et al. [41] as well as the findings of this investigation. We found that the plastid genomes of our two samples of *C. caroliniana* differ from each other by a total of 12 nucleotide changes across the aligned coding regions as well as several DNA insertions/deletions (Table 5). The species *C. caroliniana* is also peculiar in exhibiting an unusual chloroplast dimorphism [70] and a nuclear genome size considerably larger than those of other taxa in the plant order Nymphaeales [71]. Based on the complete plastid genome sequences presented here, future research may evaluate if the regions identified by McCracken et al. [41] as most variable across the plastid genome are indeed the preferred genetic markers for delimiting populations in *C. caroliniana*.

In the present investigation, we emphasized the need for standardized bioinformatic workflows in genomic research, particularly for phylogenomic studies. Given the large amounts of NGS data involved in genomic research, the process of quality filtering, assembling and annotating genome datasets cannot be done by hand but requires automated and computerized workflows [72,73]. Thus, genomic research requires the application of customized bioinformatic pipelines to channel the DNA sequence data to the proper steps of analysis. Phylogenomic investigations are in particular need for such pipelines, as they require additional analysis steps such as sequence alignment and phylogenetic tree inference, which can be extremely labor-intensive on genome-sized datasets. First attempts at constructing such pipelines exist [74,75]. A fully automated, web-based pipeline has also been developed recently [76], but it neither conducts genome assembly and assembly evaluation, nor can it be executed on the local computer of a user or customized.

Many aspects of plastid phylogenomics continue to await their integration into standardized workflows. For example, many plastid phylogenomic investigations skip sequence alignment of, and phylogenetic tree inference based on, the non-coding, AT-rich sections of complete plastid genomes [4,7], often due to challenges in confirming sequence and alignment validity of these regions [8]. However, a high level of comprehensiveness throughout all sections of a plastid genome (i.e., coding and non-coding) is fundamental, not least in investigations at or below the species level.

Central to the present investigation is a detailed account of our bioinformatic steps for genome assembly and annotation, the extraction and alignment of the coding regions, and phylogenetic tree inference. We provide eleven software scripts to automate these bioinformatic steps as well as the generation of various assembly quality statistics. We also provide a tutorial (Appendix A) that assists users to replicate our analyses step by step. The design of our workflow was governed by the idea of standardization [34], reproducibility [33,77] and workflow extensibility [78]. Specifically, we aimed to provide a simple, stable and component-based set of software scripts that allows subsequent researchers to replicate our analyses [79] and customize the workflow [78]. All scripts were written for UNIX or platform-independent interpreters, such as Bash, Python 2.7 or R 3.4, and their usage was designed to be command-line driven to allow for easy automation [80].

## 5. Conclusions

Genomic research requires the application of customized bioinformatic workflows and the precise documentation of analysis settings, as the choice of software and parameters can affect downstream analyses and, ultimately, the final results. Currently available software tools for plastid phylogenomics have not been sufficiently integrated into workflows to automate many of the processes involved. With the present investigation, we aim to supplement currently available software tools for plastid phylogenomics by sharing a set of simple, component-based scripts that assist in assembling, annotating and quality-checking complete plastid genomes as well as subsequent alignment and phylogenetic tree inference. By making these scripts widely accessible, we hope to support the standardized application and the dissemination of bioinformatics workflows for plastid phylogenomics.

## Figures and Tables

**Figure 1 life-08-00025-f001:**
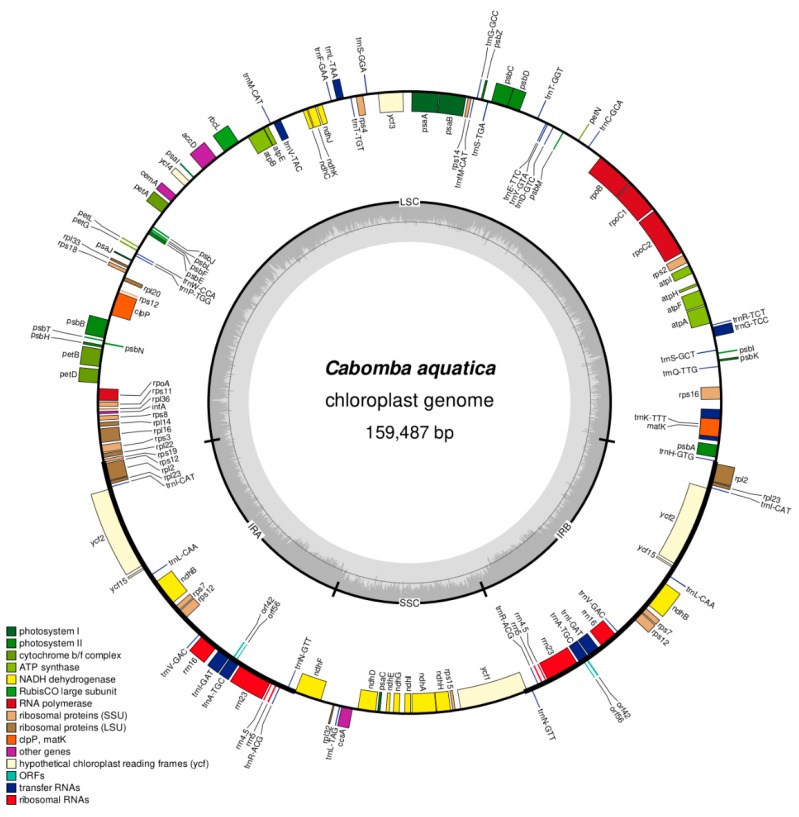
Circular plastid genome map of *Cabomba aquatica* (MG720559). Genes depicted as facing inward from the outer circle are transcribed clockwise, those facing outward are transcribed counter-clockwise. The inner circle indicates the GC content of each nucleotide position (dark gray).

**Figure 2 life-08-00025-f002:**
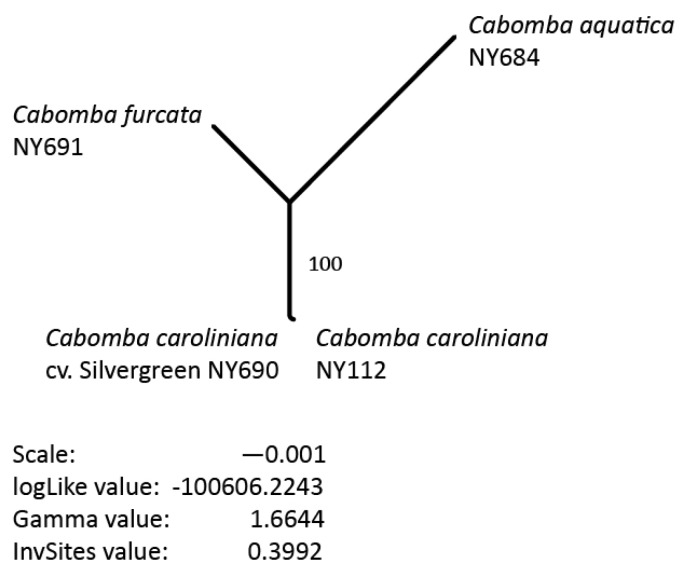
The best phylogenetic tree inferred under the ML criterion from the gap-free alignment visualized as unrooted phylogram. Bootstrap support for the inferred nodes is given as branch label. A branch length scale, the log-likelihood value, the gamma distribution value and the value for the invariant sites parameter of the best ML tree are given below the tree.

**Table 1 life-08-00025-t001:** List of species names, herbarium vouchers and GenBank accession number of each taxon analyzed. The taxonomic authority of each described species is given after the specific epithet. Standard herbarium abbreviations are given in parentheses.

Species Name	Publication of Genome Sequence	GenBank Accession Number	Sample ID	Herbarium Voucher
*C. aquatica* Aubl.	this study	MG720559	NY684	Gartenherbarbeleg Cubr 50791 (B)
*C. caroliniana* A. Gray	Gruenstaeudl et al. (2017)	KT705317	NY112	J.C. Ludwig s.n. (VPI)
*C. caroliniana* cv. ‘Silvergreen’	this study	MG720558	NY690	Gartenherbarbeleg Cubr 50793 (B)
*C. furcata* Gardner	this study	MG967470	NY691	Gartenherbarbeleg Cubr 50792 (B)

**Table 2 life-08-00025-t002:** Overview of the bioinformatic workflow applied, including the use each custom software script, names and version numbers of third-party software tools employed or initiated by the scripts, and the computation time required to perform each automated analysis step. Computation time was measured on a machine with an i5-2500K 3.3 GHz Intel Quad-Core processor (Santa Clara, CA, USA), 24 GB of RAM and a Linux 4.14.1 kernel and is given as the average computation time for each of the three newly sequenced plastid genomes. Abbreviations: min = minutes; n.a. = not applicable; s = seconds.

Analysis Step	Details of Analysis	Custom Software Script	Third-Party Software Tool Employed ^1^	Computation Time (Mean)
Quality control of reads	Generating ordered intersection of R1/R2 reads	Script 01	bioawk v.20110810	6 min 29 s
	Filtering of reads by quality score	Script 02	FASTX Toolkit v.0.0.14	14 min 03 s
Genome assembly	Assembling reads to contigs	Script 03	IOGA v.20160910 Python v.2.7.14	17 min 12 s
	Stitching contigs to complete genomes	Manual step	Geneious v.10.2.3	
Evaluation of assembly	Confirming the IR boundaries	Script 04	BLAST+ v.2.4.0	<1 s
	Extracting reads that map to final assembly	Script 05	bowtie2 v.2.3.2 samtools v.1.7 BEDtools v.2.27.1	3 min 37 s
	Generating assembly statistics	Script 06	bowtie2 v.2.3.2 samtools v.1.7 BEDtools v.2.27.1	7 min 20 s
Genome annotation	Generating raw annotations	Manual step	DOGMA cpGAVAS	
	Converting annotations of DOGMA to GFF format	Script 07	n.a.	<1 s
	Combining annotations of DOGMA and cpGAVAS	Script 08	Python v.2.7.14 Perl v.5.26.1	<1 s
Evaluation of annotations	Confirming the validity of annotations	Manual step	Geneious v.10.2.3	
Sequence alignment	Extraction and alignment of coding regions	Script 09	Python v.2.7.14	39 s
	Removal of gap positions	Script 10	statAl v.1.4.rev22 trimAl v.1.4.rev22	<1 s
Phylogenetic analysis	Phylogenetic tree inference under ML, including bootstrapping	Script 11	R v.3.4.4	17 s

^1^ Software tools such as sed, grep and awk, which are part of most Unix command shells, are not listed separately. Similarly, individual R, Python or Perl dependencies are not listed separately.

**Table 3 life-08-00025-t003:** Overview of the number of read pairs, mean coverage depth, contig number, contig length and other assembly and contig statistics. Only paired reads were counted for read statistics; orphaned reads were discarded prior to quality filtering. Contig statistics are based on contigs of a size equal to or greater than 1000 bp. Information in square brackets indicates the unit that the values are presented in. Abbreviations: N = number; P = percentage.

	*C. aquatica*	*C. caroliniana* cv. ‘Silvergreen’	*C. furcata*
N of read pairs after quality filtering	1,896,979	4,005,075	4,132,180
N of read pairs that mapped to the reference genomes (P of quality-filtered pairs) ^1^	73,452 (3.87%)	55,254 (1.37%)	46,681 (1.12%)
Mean read length [bp] ^2,3^	593.58	595.09	594.30
Mean coverage depth [fold] ^3^	349.36	266.01	224.87
P of bases with coverage depth greater than 20-fold, 50-fold, and 100-fold	99.89%—99.70%—98.51%	99.97%—99.93%—90.40%	99.87%—99.76%—89.72%
N of contigs after automatic assembly	3	4	3
Size of largest contig [bp]	89,168	80,536	90,298
Total length of contigs [bp]	134,685	135,444	135,367
N50 [bp]	89,168	80,536	90,298
L50	1	1	1

^1^ All quality filtered reads that mapped concordantly one or more times to the complete plastid genomes of *Cabomba caroliniana* and *Brasenia schreberi* were counted. Read pairs located in the IR usually map to a reference genome more than one time. ^2^ Calculated as mean length of (R1 plus R2). ^3^ Calculated from all quality-filtered reads that mapped to the final assembly.

**Table 4 life-08-00025-t004:** Comparison of genome structure, IR length and gene content of the complete plastid genomes of *Cabomba*. Abbreviations: N = number.

Name of Organism	*C. aquatica*	*C. caroliniana*	*C. caroliniana* cv. ‘Silvergreen’	*C. furcata*
Genome size (bp)	159,487	164,057	160,177	160,271
LSC length (bp)	89,433	82,090	89,835	90,037
SSC length (bp)	19,114	18,827	19,392	19,384
IR length (bp)	25,470	31,570	25,475	25,425
N of genes	116	116	116	116
N of protein-coding genes (duplicated in IR)	82 (9)	82 (19)	82 (9)	82 (9)
N of tRNA genes (duplicated in IR)	30 (7)	30 (7)	30 (7)	30 (7)
N of rRNA genes (duplicated in IR)	4 (4)	4 (4)	4 (4)	4 (4)
Proportion of coding to non-coding regions	0.69	0.69	0.69	0.69
Average gene density (genes/kb)	0.85	0.89	0.85	0.85
GC content (%)	38.0	38.3	38.0	38.1

**Table 5 life-08-00025-t005:** Overview of alignment statistics calculated on the concatenation of aligned coding regions. Abbreviations: N = number; nucl. pos. = nucleotide positions; P = percentage.

Statistic	Before Gap Removal	After Gap Removal
Total alignment length (bp)	68,922	68,451
Average pairwise sequence identity across *Cabomba* ^1^ (Pairwise sequence identity of *C. caroliniana*)	0.9852 (0.9986)	0.9891 (0.9998)

^1^ Calculated across all pairs of the concatenated coding regions.

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
