# Peer review of "Bioinformatic Workflows for Generating Complete Plastid Genome Sequences—An Example from Cabomba (Cabombaceae) in the Context of the Phylogenomic Analysis of the Water-Lily Clade"

_life, 2018, doi:10.3390/life8030025_

Round 1

Reviewer 1 Report

The inclusion of a detailed tutorial, ending in a phylogenetic tree, satisfies the prior concerns we had with this paper. Although some reviewers criticized the lack of a clear biological question, I think it's appropriate to publish methods papers that will assist other botanists in phylogenomics. This fits nicely within our stated desire to see "best practices" published in this special issue. 

Author Response

We thank the reviewer for his evalution of our manuscript. We look forward to seeing our manuscript as part of the special issue "Open Science Phyloinformatics: Resources, Methods, and Analyses".

Reviewer 2 Report

I insist in that the title of the manuscript should exclude the word phylogenomics because the workflow proposed could be used in any comparative omics study, not only in phylogenomics.  As well, both the introduction and Discussion sections would benefit from pruning redundant sentences.

line 44 & line47: should be united, the syntax is iterative, both sentences begin identically: "Plastid phylogenomics has (also) been employed..."

line 49: this concept has been stated above, the paragraph should begin with "Several...."

line 61-62: "the comparison of plastid genomes can help to infer phylogenetic relationships at various taxonomic levels" these repeats what has been stated in lines 43-48. Please erase.

line 84-86: I still deem that the quotation 30 does not add any important information, at least for your main point.

line 89: "and phylogenetic inference" should be added to be coherent. All these processes are highly sensitive to setting parameters (as the authors stated in line 22 and later on).

line 129 & line 144: the rationale to use solely the coding regions is lacking.

line 148: "Our description..."; line 149: "Our description..." Please refine your syntax and use more elaborate sentences. Should unite both sentences.

line 151 and line 159: here, another example: "other researchers to evaluate and replicate our analyses." the whole thing is repeated twice in the same paragraph.  

M&M section: Following manufacturer´s instruction is advocated FIVE times, I wonder if not a single modification/optimization was attempted, as no details are given... isn´t this incoherent with promoting reproducibility? At least in a general manner, they should emphasize that no modifications were needed, these information are also essential for reproducibility.

line 243 / 311: Brasenia schreberi (Cambobaceae) was used as a reference even though no pipeline is mention in Gruenstaeudl et al. PSE 2017. Neither the genomes documented in BMC genomics 2007 and in BME 2004 have associated pipelines. These are auto inconsistencies. Plus, any “manual intervention” like annotation correction (line 325) is against repeatability.

line 340-342: please rephrase, it says the same that 336-338.

342-344:  the authors did not explain the rationale to use solely the coding regions among 4 species of the same genus (!).

346- 351:  proceedings are repeated (i.e., "translated  from  nucleotides  to  amino  acids"). Mafft settings should be clearly indicated, to be coherent.

375: Phylogenetic analysis: there is no reason to perform such an analysis with 4 species. It is clear to me that this was done to add some biological issue. In addition, and to illustrate my point, the authors dedicated 13 seconds for analysis. Still, the biological question or meaning is lacking. The authors could have use their own dataset on Nymphaeales (15 Gruenstaeudl et al 2017), BUT as this is not the main goal of the MS, they restrict to 13 seconds and 4 species.

Should indicate G, I, Lnl values; 100 bootstrap replicates is highly insufficient (13 sec). Clearly the authors have access to important computer capabilities (line 399), thus computing time should not be a limiting factor to perform a standard (1000 or more) bootstrap analysis as should be performed nowadays.

383-386 should be place somewhere else, maybe in file 11 description

395 re-phrase

 Table 3 from the comparison of v.1 and v.2 of the manuscript I realize that this table is changed, and these changes are not indicated anywhere.

470-473: single sentence, re-phrase.

 488: The scale bar is lacking in the unrooted topology of figure 2.

table 5: the idea behind "N of nucl. pos. with gap in 25%, 50% and 75% of all taxa" is at least unclear. I deem it is useless.

The maximum sequence identity refers to a value calculated for C. caroliniana concatenation of coding regions, thus "maximum" does not make sense. It’s THE pairwise value before/after gap removal. 

506-508 It is not clear if this sentence is based on a pairwise comparison or if this was deduced from the alignment of the 4 species, as it refers to table 5.

509-510: trivial

555-581: stated before, please reduce repetition of concepts. Believe in the understanding of the reader!!!

Discussion: this section is still overemphasizing the point: lines 492-516: kind of biological discussion / 517-524 + 548-581: virtues of the pipeline and scripts.

This work is a purely methodological description of bioinformatic procedures. For this reason I would suggest its submission to the MDPI journal “Data”, which scope states that it is “a forum to publish methodical papers on processes applied to data collection, treatment and analysis, as well as for data descriptors publishing descriptions of a linked dataset” . Alternatively, MDPI Bioinformatics, Methods and Protocols, or even Genes, are other more suitable options.

Author Response

We thank the reviewer for his/her evaluation of our manuscript. We diligently revised the manuscript to incorporate nearly all of the advised changes.

COMMENT: I insist in that the title of the manuscript should exclude the word phylogenomics because the workflow proposed could be used in any comparative omics study, not only in phylogenomics.
  RESPONSE: We understand the objection by the reviewer. Thus, we rephrased the title so that it becomes clear that the workflow could be used in any comparative omics study, while mentioning that it was tested in a phylogenomic context.

COMMENT: As well, both the introduction and Discussion sections would benefit from pruning redundant sentences.
  RESPONSE: Done as requested: Redundant sentences pruned from Introduction and Discussion.

COMMENT: line 44 & line47: should be united, the syntax is iterative, both sentences begin identically: "Plastid phylogenomics has (also) been employed..."
  RESPONSE: Done as requested: Style adjusted to be less repetitive.

COMMENT: line 49: this concept has been stated above, the paragraph should begin with "Several...."
  RESPONSE: Done as requested: First sentence removed.

COMMENT: line 61-62: "the comparison of plastid genomes can help to infer phylogenetic relationships at various taxonomic levels" these repeats what has been stated in lines 43-48. Please erase.
  RESPONSE: Due to reductions in lines 43-48, the information in lines 61-62 is no longer redundant.

COMMENT: line 84-86: I still deem that the quotation 30 does not add any important information, at least for your main point.
  RESPONSE: While the quotation itself is valuable, as it nicely demonstrates the anticipated growth of plastid genomes in GenBank over the coming years, we understand that our point may have been overemphasized; the paragraph was, thus, adjusted accordingly.

COMMENT: line 89: "and phylogenetic inference" should be added to be coherent. All these processes are highly sensitive to setting parameters (as the authors stated in line 22 and later on).
  RESPONSE: Done as requested.

COMMENT: line 129 & line 144: the rationale to use solely the coding regions is lacking.
  RESPONSE: Done as requested: Rationale is inserted.

COMMENT: line 148: "Our description..."; line 149: "Our description..." Please refine your syntax and use more elaborate sentences. Should unite both sentences.
  RESPONSE: Done as requested.

COMMENT: line 151 and line 159: here, another example: "other researchers to evaluate and replicate our analyses." the whole thing is repeated twice in the same paragraph.
  RESPONSE: Adjusted as requested.

COMMENT: M&M section: Following manufacturer´s instruction is advocated FIVE times, I wonder if not a single modification/optimization was attempted, as no details are given... isn´t this incoherent with promoting reproducibility? At least in a general manner, they should emphasize that no modifications were needed, these information are also essential for reproducibility.
  RESPONSE: Done as requested: Style adjusted to be less repetitive.

COMMENT: line 243 / 311: Brasenia schreberi (Cambobaceae) was used as a reference even though no pipeline is mention in Gruenstaeudl et al. PSE 2017. Neither the genomes documented in BMC genomics 2007 and in BME 2004 have associated pipelines. These are auto inconsistencies.
  RESPONSE: The complete plastid genomes of Cabomba caroliniana and Brasenia schreberi had been de-novo assembled in Gruenstaeudl et al. (2017) using the same set of software tools.

COMMENT: Plus, any “manual intervention” like annotation correction (line 325) is against repeatability.
  RESPONSE: Virtually all bioinformatic pipelines/workflows require a manual intervention at some point. This must not stop science from developing/publishing workflows that strive to automate as many steps as possible and make them as repeatable as possible.

COMMENT: line 340-342: please rephrase, it says the same that 336-338.
  RESPONSE: Done as requested: Rephrased/reduced to be less repetitive.

COMMENT: 342-344:  the authors did not explain the rationale to use solely the coding regions among 4 species of the same genus (!).
  RESPONSE: Rationale has already been inserted in lines 127-129.

COMMENT: 346- 351: proceedings are repeated (i.e., "translated from nucleotides to amino acids"). Mafft settings should be clearly indicated, to be coherent.
  RESPONSE: Done as requested: Rephrased/reduced to be less repetitive; MAFFT settings added.

COMMENT: 375: Phylogenetic analysis: there is no reason to perform such an analysis with 4 species. It is clear to me that this was done to add some biological issue. In addition, and to illustrate my point, the authors dedicated 13 seconds for analysis. Still, the biological question or meaning is lacking. The authors could have use their own dataset on Nymphaeales (15 Gruenstaeudl et al 2017), BUT as this is not the main goal of the MS, they restrict to 13 seconds and 4 species.
  RESPONSE: This comment criticizes us for an analysis that was done as a proof-of-concept and which was requested by one of the other reviewers (and the special issue editor). Please see from their latest review report: “The inclusion of a detailed tutorial, ending in a phylogenetic tree, satisfies the prior concerns we had with this paper. Although some reviewers criticized the lack of a clear biological question, I think it's appropriate to publish methods papers that will assist other botanists in phylogenomics. This fits nicely within our stated desire to see "best practices" published in this special issue.”

As for the analysis time of 13 seconds: An equivalent maximum likelihood with analysis using the software RAxML v.8.2.9 on the same dataset takes less than a second (0.43 secs). The duration of the analysis is not indicative for its quality.

COMMENT: Should indicate G, I, Lnl values; 100 bootstrap replicates is highly insufficient (13 sec). Clearly the authors have access to important computer capabilities (line 399), thus computing time should not be a limiting factor to perform a standard (1000 or more) bootstrap analysis as should be performed nowadays.
  RESPONSE: Done as requested: A new phylogenetic analysis was conducted with 1000 bootstrap replicates; Figure 2 and the text were adjusted appropriately. Moreover, logLike, gamma and invariant-sites values were added to plot of ML tree. The software script for phylogenetic analysis (i.e., script #11), which is co-supplied with the manuscript, was amended accordingly, as was Table 2.

COMMENT: 383-386 should be place somewhere else, maybe in file 11 description
  RESPONSE: There must be a confusion, as lines 383-386 are the footnote to Table 2 in our document view.

COMMENT: 395 re-phrase
  RESPONSE: Done as requested: Paragraph rephrased/reduced.

COMMENT: Table 3 from the comparison of v.1 and v.2 of the manuscript I realize that this table is changed, and these changes are not indicated anywhere.
  RESPONSE: The update in Table 3 from manuscript version life-282651 to version life-305093 was only technical in nature. It is the result of including additional reads for C. aquatica and minor updates to how the assembly statistics were calculated (e.g., read pairs were counted instead of all reads). These changes did not go beyond Table 3 itself.

COMMENT: 470-473: single sentence, re-phrase.
  RESPONSE: Done as requested: Re-phrased sentence.

COMMENT: 488: The scale bar is lacking in the unrooted topology of figure 2.
  RESPONSE: Done as requested: A scalebar was added to Figure 2. The software script for phylogenetic analysis (i.e., script #11), which is co-supplied with the manuscript, was amended accordingly.

COMMENT: Table 5: the idea behind "N of nucl. pos. with gap in 25%, 50% and 75% of all taxa" is at least unclear. I deem it is useless.
  RESPONSE: We agree. Thanks for pointing this out. While these statistics can be helpful in estimating by how many taxa of a dataset indels are shared, they are indeed of little use in the current context and were, thus, removed from the manuscript.

COMMENT: The maximum sequence identity refers to a value calculated for C. caroliniana concatenation of coding regions, thus "maximum" does not make sense. It’s THE pairwise value before/after gap removal.
  RESPONSE: We agree. Thanks for pointing this out. Adjusted as requested.

COMMENT: 506-508 It is not clear if this sentence is based on a pairwise comparison or if this was deduced from the alignment of the 4 species, as it refers to table 5.
  RESPONSE: Done as requested: Sentence rephrased/reduced.

COMMENT: 509-510: trivial
  RESPONSE: Done as indicated: Sentence removed.

COMMENT: 555-581: stated before, please reduce repetition of concepts. Believe in the understanding of the reader!!!
  RESPONSE: We agree. The issue was resolved by removing redundant sentences in the Discussion and the Conclusion sections.

COMMENT: Discussion: this section is still overemphasizing the point: lines 492-516: kind of biological discussion / 517-524 + 548-581: virtues of the pipeline and scripts.
  RESPONSE: We agree. The issue was resolved by reducing and removing sentences in the Discussion section.

COMMENT: This work is a purely methodological description of bioinformatic procedures. For this reason I would suggest its submission to the MDPI journal “Data”, which scope states that it is “a forum to publish methodical papers on processes applied to data collection, treatment and analysis, as well as for data descriptors publishing descriptions of a linked dataset”. Alternatively, MDPI Bioinformatics, Methods and Protocols, or even Genes, are other more suitable options.
  RESPONSE: Our manuscript has a strong focus on methodological aspects because that is exactly what the special issue „Open Science Phyloinformatics: Resources, Methods, and Analyses“ requests. According to the special issue announcement, the goals of this issue are „to assemble articles on phyloinformatic resources, tools, and research reports that exemplify best practices supporting open science reproducibility and/or demonstrate inference of broad-scale patterns“. Our manuscript is, thus, appropriate for this special issue of the journal Life.